# The Influence of the Welding Process on the Ultrasonic Inspection of 9%Ni Steel Pipe Circumferential Welded Joints

**DOI:** 10.3390/ma13040961

**Published:** 2020-02-21

**Authors:** João da Cruz Payão Filho, Elisa Kimus Dias Passos, Rodrigo Stohler Gonzaga, Daniel Drumond Santos, Vinicius Pereira Maia, Diego Russo Juliano

**Affiliations:** 1Programa de Engenharia Metalúrgica e de Materiais, Instituto Alberto Luiz Coimbra de Pós-Graduação e Pesquisa de Engenharia, Universidade Federal do Rio de Janeiro (PEMM/COPPE/UFRJ), Cidade Universitária, Ilha do Fundão, Caixa Postal 68505, CEP 21941-972 Rio de Janeiro—RJ, Brazil; jpayao@metalmat.ufrj.br (J.d.C.P.F.); rodrigovr@metalmat.ufrj.br (R.S.G.); 2Serviço Nacional de Aprendizagem Industrial do Rio de Janeiro, Instituto Senai de Tecnologia de Solda (IST Senai Solda), R. São Francisco Xavier, no 601, Maracanã, CEP 20550-011 Rio de Janeiro—RJ, Brazil; drumondsantos@gmail.com (D.D.S.); vinicius.maia00@gmail.com (V.P.M.); 3Shell Petróleo Brasil Ltd.a., Av. República do Chile, no 330, 25° floor, Torre Oeste, Centro, CEP 20031-170 Rio de Janeiro—RJ, Brazil; diego.juliano@shell.com

**Keywords:** nondestructive tests, dissimilar weldments, phased array ultrasonic tests, attenuations, computer simulations

## Abstract

This work aims to compare the ultrasonic inspection of 9%Ni steel joints welded with the Gas Metal Arc Welding (GMAW) process and Shielded Metal Arc Welding (SMAW) process. These are the two most widely used processes used to weld pipes for CO_2_ injection units for floating production storage and offloading (FPSO) in the Brazilian oil and gas industry. The SMAW equipment is simple and portable, which is convenient for the FPSO; however, the GMAW process has the advantage of welding with high productivity. In this study we performed a numerical simulation using the software CIVA, 11th version, to analyze the behavior of ultrasonic longitudinal wave beams through GMAW and SMAW dissimilar weld joints. Ultrasonic tests were performed on calibration blocks drawn from both welded joints to evaluate the simulation results. The results are discussed with regard to the microstructure of the weld metal via electron backscatter diffraction (EBSD) analyses. The SMAW process presented better inspection performance than the GMAW process in terms of attenuation and dispersion effects. Although the SMAW had a better outcome, for both processes the configuration of 16 active elements and a scanning angle of 48° resulted in an optimized inspection of the entire joint.

## 1. Introduction

Since 2010, 9%Ni steel has been used in CO_2_ injection units in Brazil’s pre-salt oil and gas reservoirs. This application is new worldwide and requires massive, thick-walled pipes because of the high working pressure in the floating production storage and offloading (FPSO) CO_2_ injection unit (around 550 bar/55 MPa). To provide joints that support the internal pressure of the pipes, a high-productivity welding process, such as gas metal arc welding (GMAW), should be used. As the most-used welding processes for 9%Ni steel in Brazil are shielded metal arc welding (SMAW) and shielded metal arc welding (GMAW), in this article we aim to compare the behavior of an ultrasonic beam projected through joints welded with these two processes.

According to Tenge and his coworkers [1], lack of sidewall fusion is the defect type that constitutes the highest degree of severity found in 9%Ni steel joints welded with Ni-based superalloy as a filler metal due to higher sensitivity to variations in welding conditions, including the groove setup. This defect is more frequent in GMAW weldments. Therefore, this technique requires higher care with cleanliness and groove setup. Other defects that may come up during the welding process are hot cracks (solidification and liquation cracks), lack of penetration, and porosity. The lack of sidewall fusion is the most critical defect because the groove face is flat and orthogonally oriented in the direction of the highest mechanical stress. Moreover, this region is adjacent to the heat-affected zone (HAZ), where the fracture toughness is usually less than that of the base metal, and there is a concentration of residual stresses produced by the thermal welding cycles.

When this internal planar defect is perpendicular to the ultrasonic wave incidence, detecting it with ultrasound is easier than identifying other defect types with the same size or larger aspect ratio [2]. This fact is explained by the sound reflected with minimal dispersion in well-oriented planar defects, which does not occur in a curved defect interface. In this last case, scattering of the reflected sound is likely to happen, and only a small portion of the emitted wave comes back to the transducer.

The anisotropic grain structure of Ni-based weld metal promotes high wave deflection and attenuation, which influences defect detection and size measurement. One of the factors that causes wave attenuation is the ultrasound scattering, influenced by the grain size, shape, orientation, distribution, and anisotropy [3]. Previous studies have shown the influence of welding processes, and their characteristics, in the grain anisotropy of austenitic weld metals [4,5,6,7].

In a pioneering work on this theme, Tomlinson and collaborators [8] observed quite different behavior of ultrasound waves in GMAW and SMAW austenitic weld metals. While the latter is characterized by a high degree of grain alignment, in the GMAW weld metal, the epitaxial growth is only perceived occasionally. The epitaxial grain-growing across many weld beads with well-defined orientation leads to a tolerable level of attenuation for a longitudinal wave sonic beam [9]. This fact is attributed to the higher and more localized heat input, which causes a deeper re-melting of underlying beads during the GMAW welding process. Tomlinson and collaborators [8] concluded that the grain growth direction in GMAW weld metal was not as well-defined as in SMAW weld metal. Instead, there was a significant level of anisotropy, combined with larger grains due to the higher heat input.

In this context, investigations were carried out to foresee ultrasonic behavior during the inspection of thick-walled 9%Ni steel pipe joints. Those joints were girth-welded with Ni-based superalloy as filler metal via GMAW and SMAW processes, being the root of both joints welded with gas tungsten arc welding (GTAW). This work is a continuation of a previous study (Payão Filho et al. [10]). To corroborate the previous results, a numerical simulation study, using the software CIVA, 11th version from Extende^®^ Company, was conducted to accurately compare the ultrasonic inspection of 9%Ni steel joints welded with GMAW and SMAW.

## 2. Materials and Methods

### 2.1. Materials

A quenched and tempered 9%Ni steel pipe, as recommended by ASTM (American Society for Testing and Materials) A333 grade 8, was used to carry out this research. The pipe had 8″ (219.1 mm) nominal outside diameter and 1¼″ (31.7 mm) thickness. The filler metals chosen were two Ni-based superalloys 625 (AWS ERNiCrMo-3 for GTAW and GMAW, and AWS ENiCrMo-3 for SMAW). The specified and analyzed mechanical properties of the base metal and the specified mechanical properties of the filler metals are shown in Table 1. The specified chemical compositions of the base and the specified and analyzed chemical compositions of the filler and weld metals are shown in Table 2.

Figure 1 shows the micrographs of the 9%Ni steel base metal using optical and scanning electron microscope (SEM) analysis. As 9%Ni steel underwent a tempering treatment between 565 °C and 605 °C (the inter-critical region), the cementite and austenite phases precipitated competitively, both acting as a substrate to nucleate one another. After the quenching treatment, a certain amount of austenite remains in the microstructure, and during the tempering treatment, cementite precipitates and is absorbed by the austenite phase. In Figure 1a, it is possible to observe dark particles at the grain boundaries (indicated by arrows), which are likely to be cementite precipitates and austenite grains. When the same region is analyzed by SEM, the cementite and austenite are indicated by light gray particles (Figure 1b–d).

### 2.2. Methods

#### 2.2.1. Welding

Two joints were used to develop this work, of which one was welded with the GMAW process (filler and cap beads) and the second using the SMAW process (filler and cap beads). The root and the hot passes of both joints were welded with the GTAW process. Figure 2 shows the joint setup (single-V with V root) and the dimensions of the 9%Ni steel pipe. The whole weld process was monitored with data acquisition equipment (IMC SAP 4.0 system), and the acquired welding parameters are presented in Table 3 [10].

Figure 3 shows the average heat input of each weld bead. The heat input was calculated considering the average current, the voltage, and travel time for each bead. Compared to the GMAW weld metal, the SMAW weld metal presents a higher number of deposited beads due to the more in-depth grinding required after each pass to ensure the absence of slag and defects promoted by the magnetic blowing. The colors represent the intensity of the heat input applied: the more reddish, the higher the heat input. The heat input is the major factor responsible for the grain size determination. Comparing the processes involved, GMAW implies a higher heat input, which promotes larger grains in the weld metal.

#### 2.2.2. Ultrasonic Tests

The ultrasonic tests were performed in two steps: the first one was a preliminary test with a conventional ultrasonic dual element transducer MSEB 4 type and 4 MHz from GE Company, to measure the attenuation of the wave inside the base metal and the GMAW and SMAW weld metals. The second one was a simulation validation test with an Omniscan MX2 32-128 phased array device, SA32-N60L-IHC type wedge, and 2.25L32-A32 phased array transducer (32 elements of capacity) and 2.25 MHz frequency (all from Olympus) [14]. The calibration of the wave speed, wedge delay, and sensibility was performed according to the configuration cited above; after that, the time-corrected gain (TCG) was traced. The transducer was positioned over the calibration blocks, withdrawn from the GTAW/GMAW and GTAW/SMAW welded joints with the weld cap flush and three through-holes of 2.5 mm diameter aligned at the fusion line, as recommended by ASME Section V, article 4 (2015). The third hole of this block was inspected without the gain compensation attained by TCG calibration. The tests were performed using the best configurations defined by CIVA simulation with regard to beam attenuation and beam coverage, i.e., 16 active elements and scanning angles of 45°, 48°, and 52°.

As recommended by the United Kingdom Atomic Energy Authority Northern Division [15], the use of low-frequency probes (1 to 2.25 MHz) minimizes the noise level created by the austenitic weld metal. Papadakis reported that the wave backscattering noise from the polycrystalline structure of the metal increases with frequency and grain diameter, so the application of a low-frequency transducer is advantageous [16].

#### 2.2.3. Computational Simulation Procedure

Based on the Ogilvy model, which is a semi-analytical model for ultrasound wave propagation, the software CIVA, 11th version (Extende, Massy, France), was used to determine the best combination of phased array ultrasonic parameters considering the influence of the material properties on the ultrasonic beam [14]. The following parameters were defined to perform the simulation.

1. Material and Geometry

The specimen geometry considered in the simulations was a rectangular block with dimensions of approximately 300 × 30 × 30 mm. The weldment is located in the center of the specimen. The simulation was performed in 2D and considered the wave propagation through the plane highlighted by orange lines in Figure 4.

In order to analyze the anisotropy level, a macrography analysis was performed for each joint. The weld metal was split into anisotropic homogeneous domains, and the orientation of each domain was identified, as shown in Figure 5. The base metal and the root region were considered isotropic. That information was adopted in the simulation model. The longitudinal ultrasound speed of the austenitic weld metal was defined as 5800 m/s, and that of the carbon steel base metal was 5900 m/s. The stiffness matrix with orthotropic symmetry presented is presented in Table 4.

2. Probe, Wedge, and Focal Law

The parameters of the probe, the wedge, and the focal law adopted in the simulation modal were the same adopted in the phased array ultrasonic tests. The excitation signal of the transducer, with a frequency of 2.25 MHz and a bandwidth of 73% at −6 dB, is presented in Figure 6.

The focal laws considered in the simulation are summarized in Table 5 (scanning incidence angles of the ultrasonic waves, the number of active elements). For each configuration, the element step was defined as 1.

#### 2.2.4. Metallographic Characterization

Micrographs of the base metal and the heat-affected zone (HAZ) of the two welded joints were made using an SEM. The SEM analysis was necessary because optical microscopy does not allow a precise microstructure identification. The sample preparation followed the conventional gridding procedure, with mesh SiC paper of 100, 220, 320, 400, and 600 grit sizes and diamond polishing to 1 μm. The samples were chemically etched with Nital 2% (a solution of 2% nitric acid with 98% alcohol) for 10 s.

For a complementary result, the weld metals of the two joints were analyzed via an Quanta 450 SEM (FEI, OR, USA) equipped with a Quantax electron backscatter diffraction (EBSD) system (Bruker, Billerica, MA, USA). The analyses were performed in the center of the weld metals in the region of filler passes to avoid the influence of epitaxial grain growth close to the fusion line. The regions analyzed by the EBSD were welded with a heat input of 1.7 to 2.0 kJ/mm for the GMAW process and 1.2 to 1.5 kJ/mm for the SMAW process, as seen in Figure 3. The step size, the work distance, and the arc voltage were 2.39 μm, 15.7 mm, and 20 kV, respectively. The samples for EBSD were prepared by mechanized polishing to 0.04 μm for 1 h, using colloidal silica and no chemical etching.

## 3. Results

### 3.1. Preliminary Tests 

In preliminary tests, the longitudinal wave attenuation found was 0.098 dB/mm, 0.146 dB/mm, and 0.225 dB/mm, for the 9%Ni steel base metal, GTAW/SMAW weld metal, and GTAW/GMAW weld metal, respectively. A marked difference in attenuation level was perceived between the two weld metals and the base metal (Table 6).

Figure 7 shows the first and the second reflection of back wall echo maximized for 80% amplitude for the GMAW joint (a), SMAW joint (b), and 9%Ni steel base metal (c). The sonic beam showed good penetration through the SMAW and GMAW weld metals; however, for the GMAW, the attenuation was markedly higher. Comparing Figure 7a_2_ to Figure 7b_2_, it is possible to observe that the noise level after the second echo was much higher for the GMAW weld metal.

### 3.2. Microstructural Characterization

Figure 8 presents an SEM micrograph of the base metal for both the GMAW and SMAW joints. The microstructure is composed of a ferritic matrix, carbides, and reverse austenite along the grain boundaries.

Figure 9 exhibits SEM micrographs of the HAZ near the fusion line for both SMAW and GMAW welded joints. The images were obtained from the cap, filler, and root pass regions. In both processes, the HAZ created by the cap passes was not reheated because it underwent just one thermal welding cycle. Moreover, the base metal adjacent to the fusion line experienced complete austenitization (reaching temperatures higher than Ac3, when the ferrite begins to transform in austenite) [17], followed by fast cooling, which resulted in a microstructure consisting of lath martensite (α′) and coalesced bainite (Bc), as shown in Figure 9a,b [18,19]. The HAZ of the SMAW joint appears to have a larger fraction of coalesced bainite and less martensite (stressed), probably due to the lower cooling rates promoted by the insulator slag.

The HAZ formed by the deposition of the filler passes (Figure 9c,d), and the root and hot passes (Figure 9e,f), were influenced by more than one welding thermal cycle, which promoted the tempering of the martensitic microstructure originating in the first weld pass. Thus, for both GMAW and SMAW joints, the microstructure was composed of ferritic (*α*) with cementite (Fe_3_C) and some austenite (*γ*). Regarding the SMAW joint, the HAZ presented polygonal ferrite (PF) and, at the root of the weld, the HAZ presented a refined microstructure of ferrite and carbides.

In Figure 9f, the region formed at the interface of base metal and weld metal is called a “beach” and is formed during the solidification of the weld metal, where a portion of the 9%Ni steel moves into the weld metal during convection movement. Since the Ni-based alloy melting point is lower than that of the base metal, there is no blend between the base metal and weld metal in this region [20]. None of the constituents found in the 9%Ni steel HAZ interfered in the beam propagation because they are more equiaxial when compared with the weld metal.

Figure 10 indicates the crystal orientation map for the SMAW and GMAW joints (left and right, respectively), where the weld metals are characterized by elongated and inclined grains (following the welding direction). In addition, these grains showed small variations in internal orientation (observed by the gradual internal variation of coloration in each grain), which are correlated with the deformations that took place during the liquid–solid transformation [21].

The poles figures (PFs), shown in Figure 11, indicate that the microstructure is oriented near the easy growth direction for the metals with cubic (<100>) crystalline structure [22]. Further, the higher intensities presented by SMAW were attributed to the formation of a slag, which reduced the heat loss by convection and radiation, intensifying competitive growth and, consequently, the formation of a more oriented microstructure than the one obtained by GMAW [23]. The pulsed arc enabled the nucleation of a larger number of grains with random orientations, which results in a less-oriented microstructure and grain refinement.

### 3.3. Computational Simulation

The behavior of the simulated ultrasonic beam passing through the weld metals of the GMAW and SMAW joints is presented in Figure 12. The colors represent the attenuation of the wave throughout the material. The same parameters were set up for both joints, using longitudinal waves with direct incidence; scanning angles of 45°, 48°, and 52°; and 16, 24, and 32 active elements.

### 3.4. Ultrasonic Testing

The ultrasonic testing results with and without the TCG are presented for different inspection configurations in Figure 13, Figure 14, Figure 15, Figure 16, Figure 17 and Figure 18 through the Ascan and Sscan views while the sound was reflected in the third hole of the calibration block. The Ascan view represents the amplitude of the ultrasound echo over the block’s base, whereas the Sscan represents the amplitude of the echo through the inspection plane over the sound path. The inspection results for GMAW and SMAW, considering an incidence angle of 45° and 16 active elements, are presented in Figure 13. For this configuration the primary gains from the third hole reflection were 44.2 dB (GMAW) and 50 dB (SMAW) after maximizing the echo from the third hole to 80% of the screen height and without the TCG adjustment; after the TCG (Figure 14) the gains were 33.5 dB (GMAW) and 22.8 dB (SMAW).

For ultrasonic scanning at an angle of 48° and 16 active elements, the primary gain from the third hole was maximized to 80% of the screen height without TCG, and the gain was 37.3 dB (GMAW) and 40.1 (SMAW) (Figure 15); after TCG tracing, the primary gain was 21.4 dB (GMAW) and 22.6 dB (SMAW) (Figure 16).

Lastly, the primary gain from the third hole for ultrasonic scanning at an angle of 52° and with 16 active elements adjusted to 80% of the screen height was 40.7 dB (GMAW) and 40.8 (SMAW) (Figure 17), and after TCG tracing, the primary gain was 23.7 dB (GMAW) and 27.2 dB (SMAW) (Figure 18). Table 7 shows the primary gain values summarized before and after the TCG tracing.

The ΔdB column shows the amount of dB that needs to be compensated in the TCG adjustment. It means that the higher this value, the higher the attenuation caused by the sound path analyzed. As can be seen, the SMAW joint has a higher attenuation effect with lower incidence angles, and GMAW has a higher attenuation effect with higher incidence angles.

## 4. Discussion

### 4.1. Preliminary Tests and Characterization

In the preliminary tests, the attenuation effect observed in the weld metals of the GMAW and SMAW joints was higher than that of the base metal, due to the changes in the speed of the sonic wave caused by the anisotropic microstructure of the latter. In practice, the attenuation factor (*α*) is defined by the ratio between the amplitude difference of the first two echoes and the distance covered by the sound [24], as presented in Equation (1):(1)$$\alpha =\frac{\Delta V_{g}}{2d}$$
where*α* = attenuation factor (dB/mm);Δ*V*_g_ = difference in amplitude of the first two echoes (dB);*d* = thickness of the material (mm).

As the attenuation is directly proportional to the frequency, as mentioned by Papadakis [16], decreasing the frequency results in a decrease in the attenuation, so an inadequate frequency increase results in a much higher attenuation, which is caused in part by the coarse-grained microstructure of the metal. Kurtulmuş suggested the use of a frequency that guarantees defect detectability without compromising the signal-to-noise ratio of the inspection [25].

Comparing the weld metals of the GMAW and SMAW joints, the attenuation and the scattering effect was higher for GMAW due to its higher grain alignment degree. Tomlinson and coworkers [8] observed that the grain structure of GMAW weld metals is quite different from that of SMAW weld metals, this being attributed to the concentration of heat input and the deeper arc penetration in the GMAW process [8].

Cubic face-centered weld metals, such as Ni-based superalloy 625, present coarse grains oriented parallel to the direction of heat flow, which is approximately vertical to the weld metal centerline and perpendicular to the fusion line. This anisotropic characteristic increases the local sonic speed variation, beam deviation, and scattering at the grain boundaries, resulting in a lower signal-to-noise ratio. Nickel-based alloy weld metals are considered difficult to inspect by ultrasound. The backscattering echo generated by the columnar crystalline microstructure of the weld metal leads to a low signal-to-noise ratio and worse detectability of the defect [26]. These factors can jeopardize the interpretation of the ultrasonic signal, which reflects the loss of accuracy in the location and the sizing of defects, in addition to making it difficult to distinguish between real defects and false indications [27].

### 4.2. Computational Simulation

Through the CIVA simulations, it was possible to observe that the results for the SMAW joint were significantly better in terms of the sound pressure and attenuation effect than the results for the GMAW joint. For an angle of 45°, for example, the sonic beam generated by 16 elements covered almost all the weld metal, although the same phenomenon could not be seen for the GMAW. Although the combination of longitudinal waves and a transducer with low frequency (2.25 MHz) promotes low beam divergence due to its large wavelength, the weld metal microstructure produced by the GMAW process provided high beam deviation. 

Comparing the outputs for the number of active elements, which were 16, 24, and 32, the first number (16) demonstrated the best signal amplitude response without losing sonic pressure or the directivity of the beam for both GMAW and SMAW joints. As the number of elements increased, the weld metal microstructure caused more significant beam scattering, also evidenced by low focus and beam narrowing. These results agree with the result found by Freitas [28], where each piezoelectric element generated a sonic beam with delayed waves that interacted with each other when returning from the defect. Therefore, if the number of elements is high, the interactions consequently increase, causing more beam scattering. The wavefront generated by an ultrasonic array transducer is the result of the formation of several small, time-delayed waves generated by each piezoelectric element. The sound pressure is proportional to the number of active elements, i.e., the active area of a transducer. Therefore, to improve the beam response, it is necessary to provide more energy, increasing the number of active elements. Freitas suggested deeper penetration can be achieved by using more active elements; however, it is essential to consider the signal-to-noise ratio coming from the microstructure [28], so the focal law will be the relation between the number of elements as a function of the signal-to-noise ratio.

In general, a transducer containing 16 elements shows the best relation between the number of elements and the spacing between them, with a pitch value close to two-thirds of the wavelength of the sonic beam. This guarantees optimization of the inspection without introducing deleterious effects generated by a large number active of elements [29].

When comparing the results of the 45° and 48° incidence angles, the latter presented better results regarding ultrasonic inspection, ensuring greater coverage. At an angle of 52°, it was possible to see beam collimation near the weld cap. These results agree with Hirsekorn [9], who showed that the beam deflection was at a minimum for angles between 45° and 50° for longitudinal waves, as seen in Figure 19. The application of longitudinal waves in coarse grain and oriented microstructure materials reduces the effect of beam deflection due to their large wavelength.

### 4.3. Ultrasonic Tests

In the ultrasonic tests, a better propagation of the sonic beam was observed for the scanning angle of 48°, which resulted in the best signal amplitude response without losing sonic pressure or directivity of the beam. In this configuration, it would be necessary to apply at least three scanning indexes to ensure inspection of the entire volume of the welded joint.

All the test configurations resulted in the detection of the three holes located at the fusion line of the calibration blocks. However, some of them needed a larger gain to ensure inspection of the whole joint. The higher the gain, the more difficult the inspection. Thus, of the three scanning angles, the angle that presented the lowest primary gain from the calibration block third hole was 48°, for both GMAW and SMAW joints, with values of 37.3 dB (GMAW) and 40.1 dB (SMAW), and 21.4 dB (GMAW) and 22.6 dB (SMAW) after tracing the TCG. These results agree with the CIVA simulation, which revealed that the SMAW weld metal promoted the lowest attenuation and the highest joint coverage.

The results of the simulation and ultrasonic experiments together produced consistent conclusions regarding the inspection of GMAW and SMAW joints with phased array ultrasonic waves. In this study, both analyses agreed in finding 48° to be the best scanning angle.

## 5. Conclusions


Preliminary tests revealed that the attenuation is significantly greater in GMAW joints than in SMAW joints.The CIVA simulation showed that the results for the SMAW joint were significantly better than those for the GMAW joint in terms of sound pressure and attenuation. The sonic beam covered almost all the weld metal, although the GMAW process needed greater sonic pressure to ensure the same behavior, which compromised the signal-to-noise ratio.The configuration with 16 active elements proved to be the most effective scanning technique for SMAW and GMAW joints inspection, covering a large volume of the weld using the same combination of low frequency (2.25 MHz) and angles of 45° and 48°.Ultrasonic tests revealed that the inspection configuration with longitudinal waves and angles of 45°, 48°, and 52° allowed the detection of the three holes aligned along the fusion line of the calibration block. However, the scanning incidence angle of 48° showed the best results.The results of the longitudinal wave inspection showed an optimized parameter range; as the scanning angle moved away from this range, the signal-to-noise increased, which could jeopardize defect detection and size estimation.The experimental tests validated the simulation results, and the hypothetical assumptions were confirmed.


## Figures and Tables

**Figure 1 materials-13-00961-f001:**
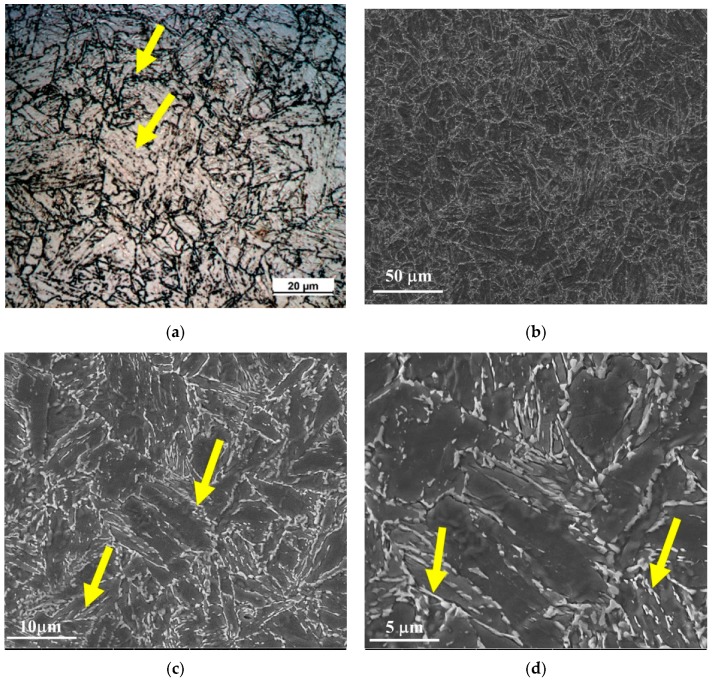
Micrographs of the 9%Ni quenched and tempered steel base metal via optical and SEM analysis with different magnifications (**a**) 500×, (**b**) 1000×, (**c**) 5000×, (**d**) 10,000×. The arrows represent precipitates of cementite and austenite.

**Figure 2 materials-13-00961-f002:**
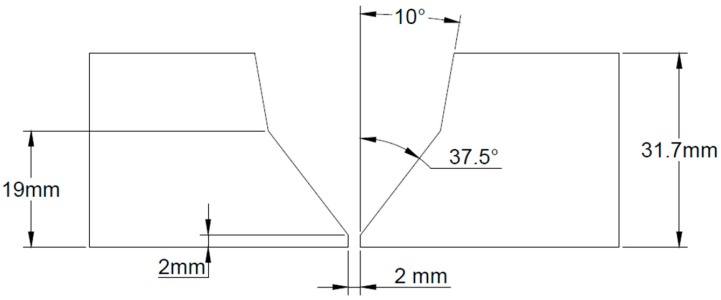
Groove preparation (single-V with V root) and dimensions of 9%Ni steel pipe.

**Figure 3 materials-13-00961-f003:**
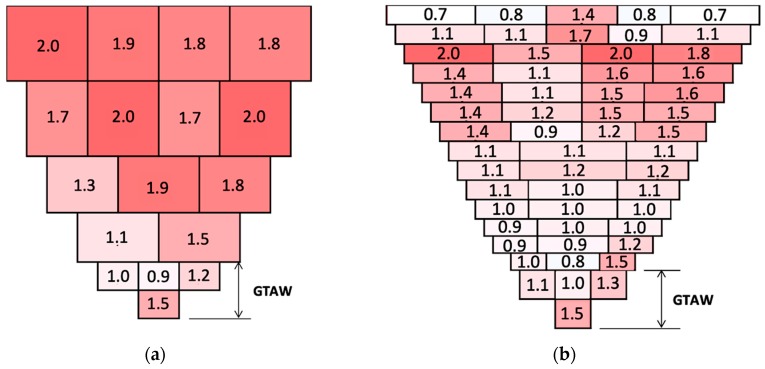
Illustration of heat input displayed by weld bead for (**a**) gas metal arc welding process (GMAW) and (**b**) shielded metal arc welding (SMAW) joints.

**Figure 4 materials-13-00961-f004:**
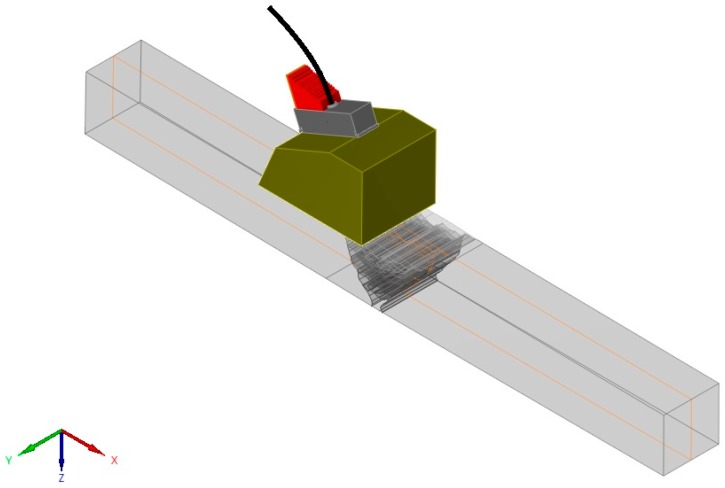
Overview of the simulated model.

**Figure 5 materials-13-00961-f005:**
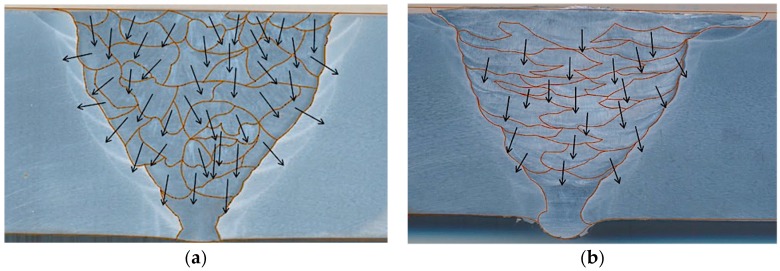
Macrography of the 9%Ni steel pipe joints girth welded with (**a**) gas metal arc and (**b**) shielded metal arc processes, highlighting the weld bead boundaries and dendrite growth directions (arrows) [13].

**Figure 6 materials-13-00961-f006:**
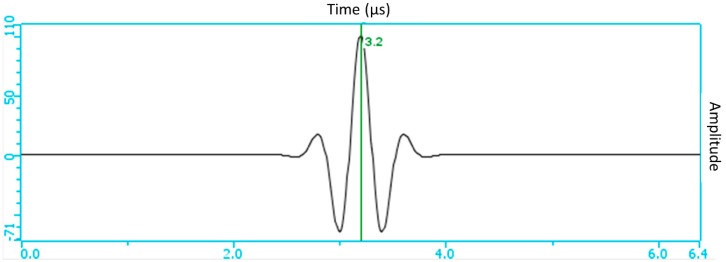
Excitation signal of the probe used in the simulation.

**Figure 7 materials-13-00961-f007:**
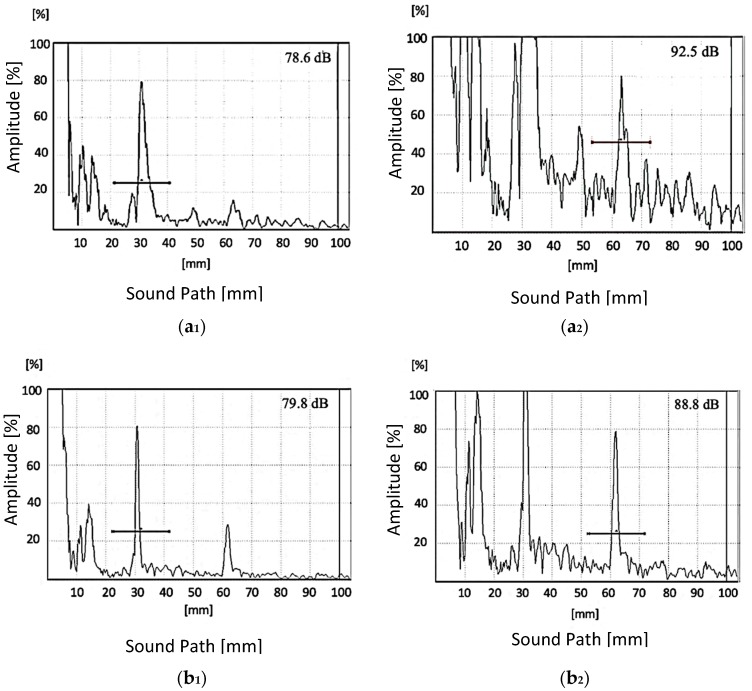
Ultrasonic reflection of background echo adjusted to 80% amplitude: GMAW (**a_1_**), SMAW (**b_1_**), and base metal (**c_1_**). The second reflection of the background echo is adjusted to 80%: GMAW (**a_2_**), SMAW (**b_2_**), and base metal (**c_2_**). The primary gain obtained is presented on the upper right-hand corner of each graphic.

**Figure 8 materials-13-00961-f008:**
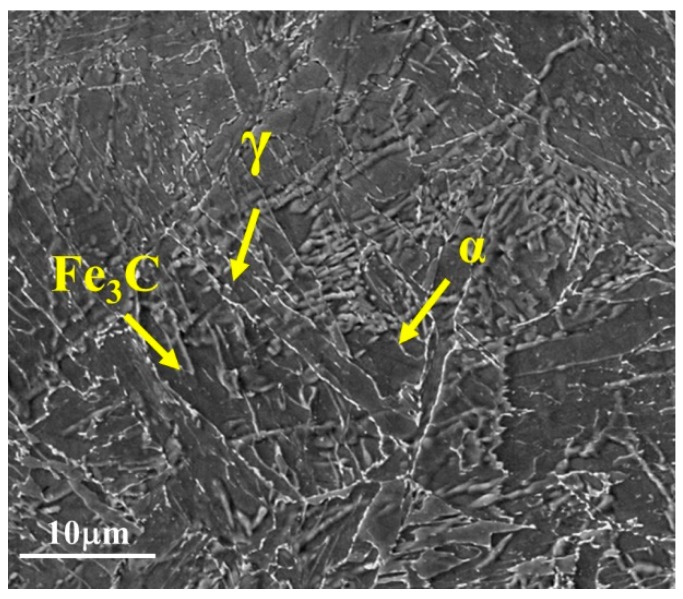
Scanning electron micrograph of the base metals of 9%Ni steel, where α is ferrite, Fe3C carbides, and γ reverse austenite.

**Figure 9 materials-13-00961-f009:**
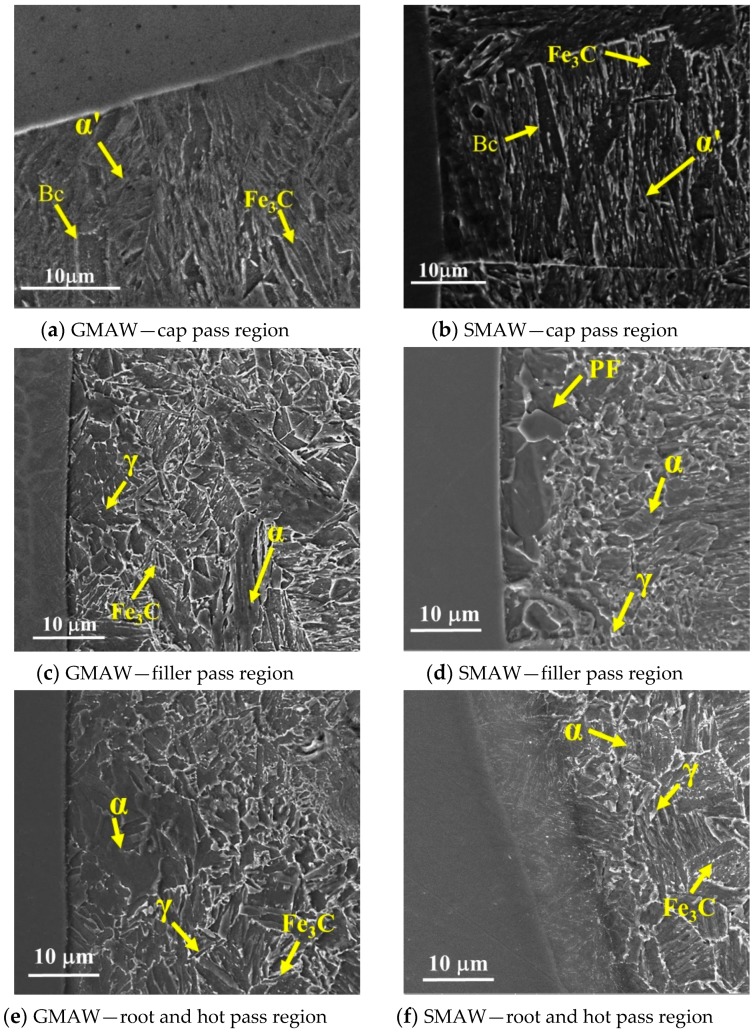
SEM micrographs of the heat-affected zone (HAZ) near the fusion line for the GMAW (**a**) cap pass region, (**c**) filler pass region, (**e**) root and hot pass region and SMAW (**b**) cap pass region, (**d**) filler pass region, (**f**) root and hot pass region. The images were obtained from the cap, filler, and root pass regions.

**Figure 10 materials-13-00961-f010:**
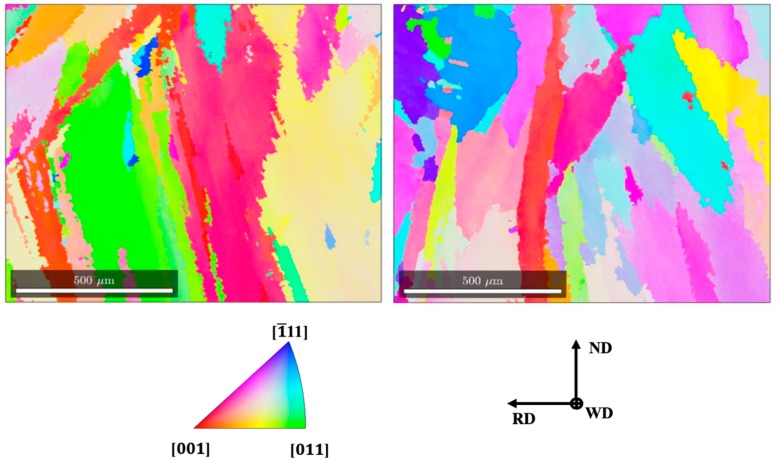
Electron backscatter diffraction (EBSD)-derived crystal orientation map for SMAW and GMAW processes [13].

**Figure 11 materials-13-00961-f011:**
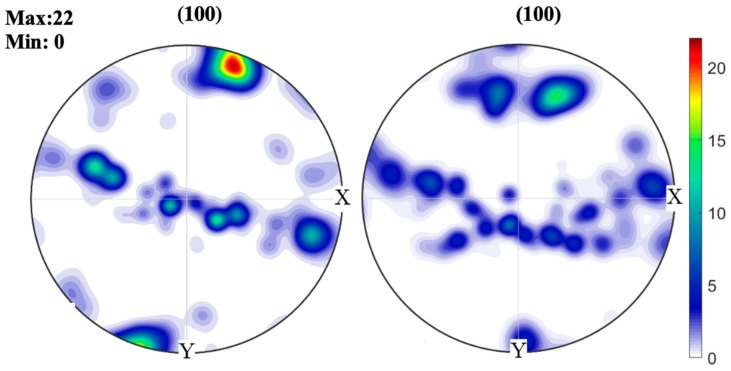
Poles figures (PFs) of the weld metal for GMAW and SMAW processes.

**Figure 12 materials-13-00961-f012:**
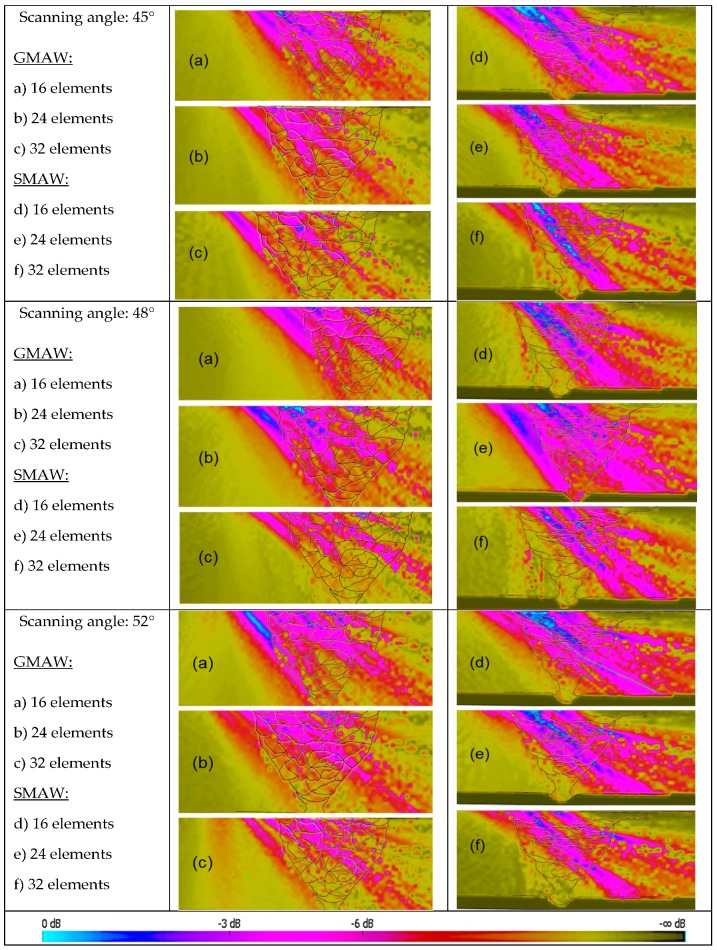
Simulation of the ultrasonic beam passing through the welded joints with scanning angles of 45°, 48°, and 52°. GMAW: (**a**) 16 active elements, (**b**) 24 active elements, and (**c**) 32 active elements; SMAW: (**d**) 16 active elements, (**e**) 24 active elements, and (**f**) 32 active elements.

**Figure 13 materials-13-00961-f013:**
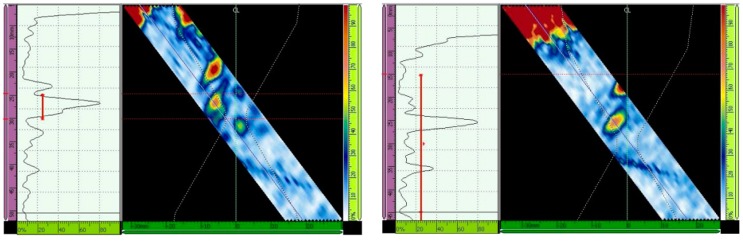
Echo (Adjusted to 80% of the screen height) from the third hole for scanning angle of 45° and 16 active elements. Primary gains were 44.2 dB (GMAW, **left**) and 50 dB (SMAW, **right**) [13].

**Figure 14 materials-13-00961-f014:**
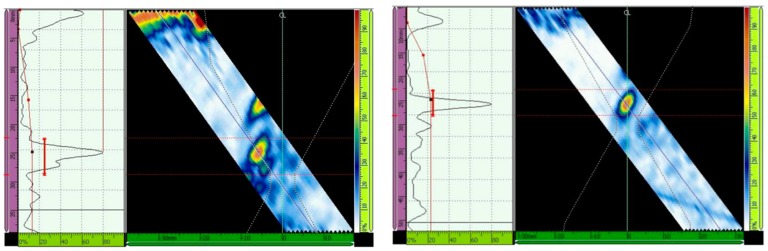
Gain adjustments for scanning angle of 45° and 16 active elements. Primary gains were 33.5 dB (GMAW, **left**) and 22.8 dB (SMAW, **right**) [13].

**Figure 15 materials-13-00961-f015:**
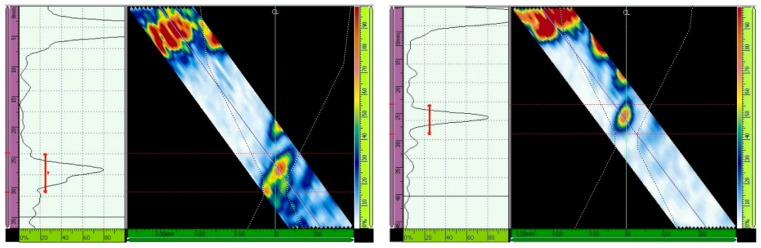
Echo (Adjusted to 80% of the screen height) from the third hole for scanning angle of 48°, 16 active elements. Primary gain = 37.3 dB (GMAW, **left**), and 40.1 dB (SMAW, **right**) [13].

**Figure 16 materials-13-00961-f016:**
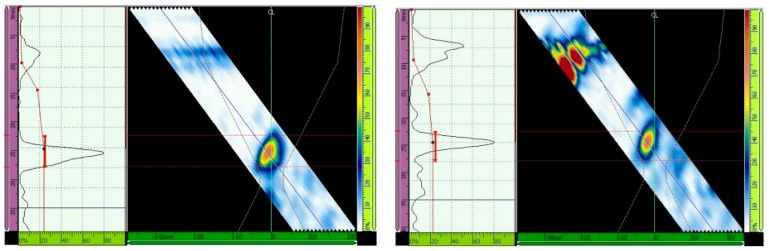
Gain adjustment for scanning angle of 48°, 16 active elements. Primary gain = 21.4 dB (GMAW, **left**) and 22.6 dB (SMAW, **right**) [13].

**Figure 17 materials-13-00961-f017:**
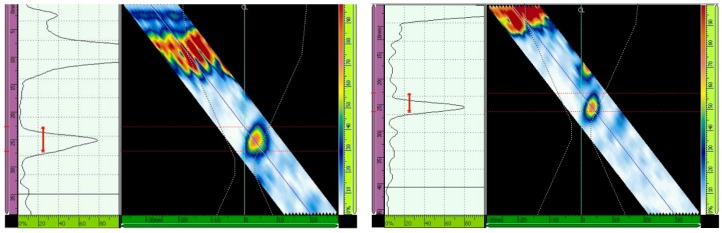
Echo (maximized to 80% of the screen height) from the third hole for scanning angle of 52°, 16 active elements. Primary gain = 40.7 dB (GMAW, **left**) and 40.8 dB (SMAW, **right**) [13].

**Figure 18 materials-13-00961-f018:**
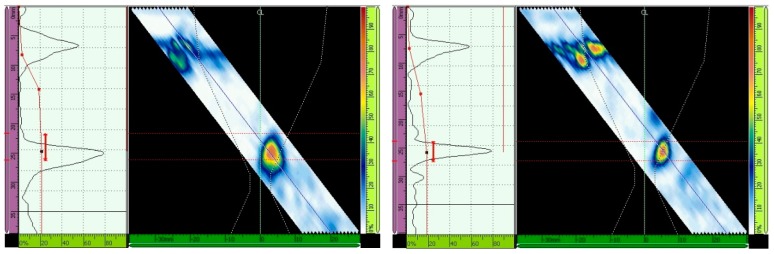
Adjustment for scanning angle of 52°, 16 active elements. Primary gain = 23.7 dB (GMAW, **left**) and 27.2 dB (SMAW, **right**) [13].

**Figure 19 materials-13-00961-f019:**
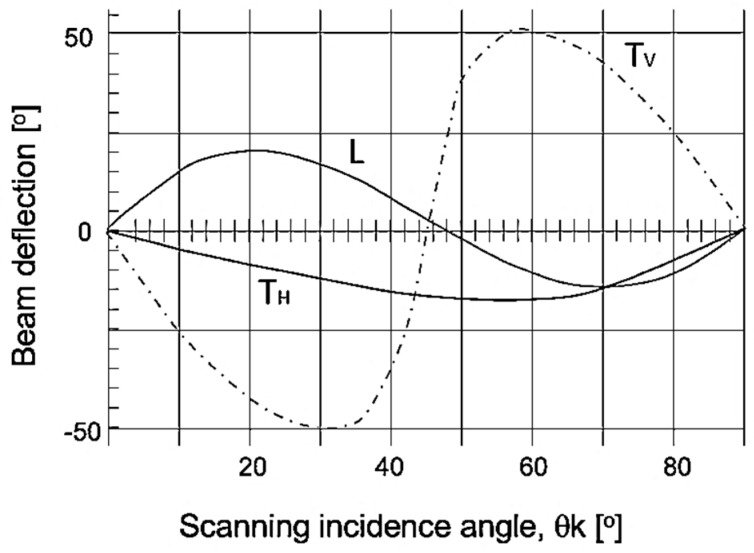
Beam deflection as a function of the scanning incidence angle (θk). T_V_ is the transverse wave with vertical polarization, T_H_ is the transverse wave with horizontal polarization, and L is the longitudinal wave [9].

**Table 1 materials-13-00961-t001:** Mechanical properties (specified by American Society for Testing and Materials, ASTM, and analyzed): yield strength (YS), ultimate tensile strength (UTS), elongation (El.), and Charpy V energy at −196 °C (CVEn), and lateral expansion (CVEx) of the base and filler metals [11,12,13].

Mechanical Property	Base Metal ASTM A333 Gr. 8 (9%Ni Steel)		Filler Metal Ni-Based Superalloy 625 Typical Value		
	Specified (min)	Analyzed	GTAW	GMAW	SMAW
YS (MPa)	515	693	510	510	530
UTS (MPa)	690	740	770	770	770
El. (%)	22	25	42	42	30
CVEn (−196 °C) (J)	N.A.	146	70	70	45
CVEx (mm)	0.38	1.78	N.A.	N.A.	N.A.

N.A. means not applicable.

**Table 2 materials-13-00961-t002:** Specified chemical composition (% in weight) of the base metal, specified and analyzed chemical compositions (% in weight) of the filler metals, as well as analyzed chemical compositions (% in weight) of the weld metals [10,11,12,13].

Element	Chemical Composition (% in Weight)								
	Base Metal	Filler Metal					Weld Metal (Analyzed)		
	Specified	GTAW/GMAW	GTAW	GMAW	SMAW		GTAW	GMAW	SMAW
		Specified	Informed by the Supplier		Specified	Informed by the Supplier			
C	0.13	<0.1	0.008	0.010	<0.1	0.0326	0.08	0.0164	0.0326
Mn	0.9	<0.5	0.02	0.02	<1.0	0.887	0.0757	0.103	0.887
Si	0.13/0.32	<0.5	0.05	0.05	<0.75	0.494	0.0784	0.102	0.494
P	0.025	<0.02	0.006	0.006	<0.03	0.0035	<0.0003	<0.0003	0.0035
S	0.025	<0.015	0.000	0.000	<0.02	0.0088	0.0009	0.0025	0.0088
Ni	8.4/9.6	>58	65.10	65.00	>55	63.1	62.9	65.0	63.1
Cu	N.A.	<0.5	0.03	0.02	<0.5	0.0061	0.0354	0.0261	0.0061
Ti	N.A.	<0.4	0.183	0.206	N.A.	0.0694	0.201	0.199	0.0694
Cr	N.A.	20/23	21.66	21.50	20/23	20	20.2	20.1	20
Mo	N.A.	8/10	8.73	8.89	8/10	8.95	8.93	9.63	8.95
Fe	Balance	<5.0	0.430	0.380	<7.0	2.49	3.49	0.840	2.49
Pb	N.A.	N.A.	0.000	0.000	N.A.	0.0504	0.0408	0.0386	0.0504
Al	N.A.	<0.4	0.140	0.180	N.A.	<0.0005	0.127	0.148	<0.0005
Nb + Ta	N.A.	3.15/4.15	3.660	3.6900	3.15/4.15	3.478	3.435	3.3170	3.478
Co	N.A.	N.A.	N.A.	N.A.	0.12	N.A.	0.0522	0.0682	N.A.
V	N.A.	N.A.	N.A.	N.A.	N.A.	N.A.	0.0345	0.0445	N.A.
W	N.A.	N.A.	N.A.	N.A.	N.A.	N.A.	0.0248	0.0343	N.A.

N.A. means not applicable.

**Table 3 materials-13-00961-t003:** Parameters adopted to girth weld the two 9%Ni steel pipe butt joints [10].

**Welding Parameter**		**Welding Process/Pass**					
		**GTAW**		**GMAW**		**SMAW**	
		**Root**	**Hot**	**Fill**	**Cap**	**Fill**	**Cap**
Current (A)		123	120	111	114	90	89
Voltage (V)		11	10	26	26	26	26
Welding speed (cm/min)		5.1	7.5	10.8	9.5	12.6	18
Average heat input (kJ/mm)		1.5	1.0	1.7	1.9	1.2	0.9
Gas type and flow rate (L/min)	Shielding	Ar */12		75%Ar + 25%He/16		N.A.	N.A.
	Purge	Ar */25		N.A.	N.A.	N.A.	N.A.
Electrode/Wire type		ERNiCrMo-3		ERNiCrMo-3		ENiCrMo-3	
Electrode/Wire diameter		2.4 mm		1.2 mm		3.25 mm	
Polarity	-	DC-		DC+		DC+	
Tip angle	-	60°		N.A.		N.A.	
Stick out	-	N.A.		15 mm		N.A.	

* 99.995% purity. N.A. means not applicable.

**Table 4 materials-13-00961-t004:** Orthotropic stiffness matrix adopted for the austenitic weld metal.

C_11_	C_12_	C_13_	C_23_	C_22_	C_33_	C_44_	C_55_	C_66_
250	112	180	138	250	250	117	91	70

**Table 5 materials-13-00961-t005:** Scanning angles of the ultrasonic waves and the number of active elements adopted in the CIVA software simulations [13].

Scanning Angle (°)	No. of Active Elements
45	16, 24, 32
48	16, 24, 32
52	16, 24, 32

**Table 6 materials-13-00961-t006:** Ultrasonic attenuation in the base metal and the SMAW and GMAW weld metals.

Region	Attenuation (dB/mm)
9%Ni steel base metal	0.098
Weld metal of the GMAW joint	0.225
Weld metal of the SMAW joint	0.146

**Table 7 materials-13-00961-t007:** Gain from the third hole and after time-corrected gain (TCG) adjustment for the different scanning angles for the GMAW and SMAW joints [13].

Scanning Angle (^o^)	Primary Gain (dB)					
	GMAW Joint			SMAW Joint		
	From the 3rd Hole	After Tracing the TCG	ΔdB	From the 3rd Hole	After Tracing the TCG	ΔdB
45	44.2	33.5	10.7	50	22.8	27.2
48	37.3	21.4	15.9	40.1	22.6	17.5
52	40.7	23.7	17	40.8	27.2	13.6

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
