# Peer review of "The Influence of the Welding Process on the Ultrasonic Inspection of 9%Ni Steel Pipe Circumferential Welded Joints"

_materials, 2020, doi:10.3390/ma13040961_

Round 1
Reviewer 1 Report
The paper uses numerical methods to evaluate the performance of the UT inspection in a welded joint. In my opinion, the topic is practical. However, there are some weaknesses that the authors must address before publication. Though the subject is interesting, the paper is not sufficient for publication in the current form. The description of test, results and conclusions are vague and questionable as well as the logic of the text.
This article is mainly to investigate the influence of weld process on the UT inspection. But in the paper, there is less information about weld process, such as torch traveling speed, gas flow rate, filler speed and so on. Mostly, what the authors presented is the post assessment of the weld, such as SEM or metallography analysis. Though I can find some processing information in table 3, it is from the literature other than the tested sample in this paper. Authors should describe the welding procedure of the samples which are used in this paper. In the title, the circumferential welded joint was mentioned. It could be really interesting if the curvature is considered in the UT inspection. However, the reported results have nothing to do with the circumferential welded joint. Why did the author list the chemical component of material (Table 2). I didn’t find the corresponding statements in the manuscript. and also, it seems that this part doesn’t really contribute to the whole paper. Why did the authors report the GTAW in table 3? The paper is to investigate the weld joint from GMAW and SWAM. Please explain how the author plot the figure 3. Is that results based on the experimental measurement or theoretical model? In addition to that, for both GMAW and SMAW results, why did the authors insert “GTAW” in the figure? Numerical simulation is as the most important part in this paper; but many details of numerical model are missing: geometric information, excitation signal, material properties, mesh, time step, types of analysis (frequency domain or transient analysis) and so on. Moreover, the proposed numerical models are not properly validated. Therefore, the reliability of the related results and analysis is low and questionable. In my opinion, this is one of the largest flaws of this article. Authors didn’t explain the results of Figure 12 in the manuscript as well as Figure 13. The quality of SEM images is low; scale bar and text are not clear. for example, in Figure 1, Figure 8, Figure 9.
Author Response
Dear Reviewer,
Thank you for your comments. We appreciate your careful assessment of our work.
This article is the continuation of another article written by our team and published in Metals Magazine, called "Ultrasonic Inspection of a 9% Ni Steel Joint Welded with Ni-based Superalloy 625: Simulation and Experimentation". Part of the information used in this work, such as Table 3 and chemical composition, has been taken from this previous article.
Some tables and references from this previous work were inserted in this new article to clarify the entire methodology adopted. Since both works had similar welding procedures, it might not be very clear in the first place. An excerpt was added in the introduction to overcome this issue.
Regarding the GTAW process, it was applied to weld the root of both SMAW and GMAW joints. It was mentioned in the text since the welding process might influence the ultrasonic inspection results. Some parts of the text were rewritten for better understanding.
When it comes to curvature, it was not considered in this work since the inspection took place in a cross-section direction of the weld. The simulation's objective was to evaluate the effect of the microstructure and not the sonic dispersion caused by the curvature of the pipe.
Figure 3 was best explained in lines 128 and 129.
Regarding numerical simulation, a text reformulation was made to clarify all the contour conditions used. New figures were performed, so that the reader can have a better understanding of the simulated problem. In the same way, the data validation was also rewritten for better understanding.
The SEM images have been corrected.
If you guess that more changes need to be made, let us know.
Best regards.
Authors
Reviewer 2 Report
The author has done pretty much effort in this work. The research appears to be efficiently done, however, the report has some failures and needs to be changed before acceptance for publication. There is some ambiguity about the explanation of results. The author needs to revise the results with appropriate citations for comparison. In introduction, the Author recommends GMAW should be chosen for high productivity but simulation shows better results for SMAW. The author should justify the contradiction. In Line 89, the Author should modify the whole sentence for clear understanding. The author should cite the reference to support the argument in fig 1 and indicate the grain separately. Correct the caption and labeling in figure 3. in reference 8, Tomlinson observed a high degree of grain alignment in SMAW than GWAM. the author did not mention the scenario in which he took these results as the Author got a high degree of grain alignment for GMAW.Author Response
Dear Reviewer,
Thank you for your comments. We appreciate your careful assessment of our work.
The result session has been changed to better explain the advantages and disadvantages of each process regarding the ultrasonic inspection. Please let us know if it meets your requirements.
In the introduction session, the citation was included to explain the general advantages of each process. This reference is focused on NDT results, which highlighted the better results of the SMAW process.
Line 89 has been modified to improve the reader's understanding.
About Tomlinson's work, he does not give many details about the inspection methodology adopted in each case, restricting himself only to signalling that one of the blocks he used was GMAW and the other SMAW. He has shown the macrography of each one. This current papaer was carried out precisely to verify this finding made by Tomlinson.
Please, let us know if we need to make any further changes.
Best regards.
Authors
Reviewer 3 Report
Review of paper titled “The influence of the welding process on the ultrasonic inspection of 9%Ni steel pipe circumferential welded joints”, by João da Cruz Payão Filho, Elisa Kimus Dias Passos, Rodrigo Stohler Gonzaga, Daniel Drumond Santos, Vinicius Pereira Maia and Diego Russo Juliano
The paper is well-written in impeccable English.
Engineering drawings should follow the drawing code specified in the latest ASME Y14.5 Dimensioning and Tolerancing version 2018 [For example, In Fig. 2 the dimension lines showing 19 mm should both point outwards, not inwards].
While the experimental ultrasonic tests and results are described in great detail, the ensuing computational sections are disproportionately small. There are no figures of the computational model, no methodology, boundary conditions etc. The mathematical formulation contains just one standard formula. More details are expected.
Given these changes are incorporated, the paper can be reconsidered for perusal for potential publication in Materials.
Author Response
Dear Reviewer,
Thank you for your comments. We appreciate your careful assessment of our work.
We have made changes in Figure 2 and in the computational section, as you suggest. I hope that it is under your expectations. If not, please, let us know.
Best regards.
Round 2
Reviewer 1 Report
The authors already addressed most of my comments. The paper can be considered to publish.
Author Response
Dear Reviewer,
Thanks for the suggestions. We appreciated your contribution.
Minor changes were made in response to your requests. The new version will be uploaded to this page.
Best regards.
Authors.
Reviewer 3 Report
The research paper is quite illustrative now; however, some language errors are now appearing in the revision - e.g., line 351 on page 18: should read
[In the ΔdB column], [t]he table shows the amount [of] dB that need[s] to [be] compensate[d] in the TCG 351 adjustment.
Besides, ASME 14.5 says that the dimensions should appear readable horizontally, not vertically, and a period should be used instead of comma in decimal numbers. Fig. 2 needs some slight corrections.
Once these are fixed, the paper is ready to be published. There is no further need for a technical review.
Author Response
Dear Reviewer,
Thanks for the suggestions. We appreciated your contribution.
The changes in Figure 2 have been made, as you suggested. They include changes in the text direction and replacement of the comma by a period. I hope the figure is now in line with your expectations.
Regarding English errors, they were revisited and corrected by the authors. Also, after technical approval, the document will be sent to an official reviewer, such as Elsevier, to improve the text format and any grammatical inconsistencies.
Best regards.
Authors.